# OpenReview forum: "The Hidden Link between RLHF and Contrastive Learning"
_ICML.cc/2026/Conference — ICML 2026 regular_

### Official Review · Reviewer_RqEV · 2026-03-12

**Soundness:** 1
**Presentation:** 1
**Significance:** 1
**Originality:** 1
**Overall Recommendation:** 2
**Confidence:** 4

**Summary:**

The paper claims that RLHF and DPO are both optimizing a Donsker-Varadhan lower bound on mutual information, which the authors argue makes them implicit forms of contrastive learning. From this framing, the authors attempt to explain why DPO fails when response probabilities are near zero, and propose MIO, which substitutes the DV estimator with a Jensen-Shannon estimator. Experiments on math and reasoning benchmarks are presented in support of the method.

**Compliance With Llm Reviewing Policy:**

Affirmed.

**Ethical Review Concerns:**

Ethics concern proved to not be an issue after discussion with program chair. The watermark looks a lot like a prompt injection. Because it is.

**Key Questions For Authors:**

How does RLVR perform on the set of benchmarks?

**Limitations:**

Yes.

**Strengths And Weaknesses:**

Soundness
The paper is broadly correct but imprecise in important places. Several essential derivations are offloaded to the appendix when they should be in the main text. The approximations in Section 2 are not always clearly labeled as such, which makes it hard to know how tight the claimed connections actually are.

The explanation for why DPO fails at low probabilities does not require the mutual information machinery the paper builds. The reason is simple: you cannot get a useful gradient signal from events your model almost never samples. This is a well-known limitation of sampling-based optimization. Reframing it in terms of the DV estimator is a valid observation, but it is not a new insight and should not be presented as one.

MIO replaces the DV estimator with JSD. This is a minor change. JSD is just a symmetrized version of KL divergence — JSD(Q,P) = ½KL(Q||M) + ½KL(P||M). The stability benefits claimed for MIO follow from this bounded, symmetric structure, which is already well understood. The paper should be upfront that this is a modest modification rather than a fundamental one.

The toy experiment in Section 4.1 is too simple to support the stability claims made. Four prompts and ten responses tell us very little about what happens when fine-tuning a 7B parameter model.

The writing contains numerous grammatical errors throughout and needs careful editing.

Presentation
The three-question structure is helpful, but the paper repeatedly makes strong claims that its derivations do not fully support. The central claim — that contrastive learning is the "secret engine" behind RLHF — is overstated. The paper shows a formal connection that is approximate and relies on restricting the critic to a specific functional family. That is worth reporting, but it is not the same as a clean equivalence.

Key arguments depend on appendix content without adequate summary in the main text. A reader should not need to read the appendix to follow the main theoretical thread.

Significance
The problem is important. Understanding why DPO fails and how to fix it matters. But the practical improvement from MIO is modest. On broader benchmarks (Table 2), gains over DPO are small and MT-Bench is identical. Math and reasoning gains look better, but the paper omits RLVR baselines, which are directly relevant given the paper's focus on reasoning failures. Without those comparisons, it is hard to know where MIO actually stands.

Originality
The contrastive learning connection to RLHF has been noted informally many times before, including in papers the authors cite. The contribution here is a more formal treatment, which has value, but the formalization is imprecise in places and the equivalence is not exact. The proposed method is a known estimator substitution applied to a new setting. That is fine, but it should be described as such.

Overall Assessment
The paper is technically sound in a basic sense — nothing is obviously wrong — but its claims are consistently larger than what the evidence supports. The theoretical connection is approximate, not exact. The gradient starvation explanation is not novel. The method is a minor modification of DPO. The empirical validation is incomplete and the toy experiment is weak. The writing needs significant cleanup.
The core idea has merit and the results are real, but the paper needs to be more honest about the scope of its contribution.

---

> ### Author Rebuttal · Authors · 2026-03-31
>
> Thank you for your time and for the careful review. However, there is some misunderstanding about our paper.
>
> ---
> **Q: Ethical Concerns**
>
> **A0**: Regarding prompt injection, please see https://icml.cc/Conferences/2026/PeerReviewFAQ#prompt_injection.
>
> ---
> **W1: Overclaims and Theoretical Scope**
>
> **A1**: Thank you for pointing this out. We agree that our original wording overstated the exactness of the connection between RLHF and contrastive learning. Our intended contribution is an approximate formal connection, not a strict mathematical equivalence.
>
> In the revision, we will tone down this framing throughout the paper, remove overly strong phrases, and move the main approximation-error result (currently in Appendix E) into the main text so that the conditions and tightness of the connection are clear without requiring the reader to consult the appendix. We have also changed the title to **“The Hidden Link between RLHF and Contrastive Learning”** to better reflect this narrower claim. We will also revise the corresponding strong statements in the abstract, contribution list, and conclusion.
>
> ---
> **W2:Clarifying the Failure Mechanism and MIO**
>
> **A2**:We agree that sampling-based optimization inherently struggles with rare events. However, our analysis reveals a deeper, DPO-specific pathology: even within a highly informative batch, as $\pi_\theta(y^-|x) \to 0$, the coupled DV bound actively suppresses the gradient for the chosen $y^+$.
>
> We also agree that replacing DV with JSD is a modest mathematical modification. Our contribution is not inventing JSD, but **demonstrating why its well-understood bounded structure specifically neutralizes this gradient starvation**. We have revised Section 3 to explicitly frame MIO as a targeted, theoretically motivated fix rather than a fundamental paradigm shift.
>
> ---
> **W3:The toy experiment is too simple**
>
> **A3**:To address this, we conducted a rigorous 1-step local update experiment on Qwen-2.5-7B-Instruct (UltraFeedback) to isolate the effect of $\log \pi_\theta(y^-|x)$. Controlling for sequence length, we stratified responses into 10 length-balanced buckets based on $\log \pi_\theta(y^-|x)$ and performed a single DPO or MIO update per bucket. We tracked:
> - rejected_sum_mean: Average $\log \pi_\theta(y^-|x)$ (lower = more unlikely).
> - grad_coeff: Gradient weight driving the $y^+$ update.
> - Δchosen_sum: The actual change in $y^+$ log probability after the update.
>
> |Bucket|rejected_sum_mean|DPOgrad_coeff|DPOΔchosen_sum|MIOgrad_coeff|MIOΔchosen_sum|
> |:---:|:---:|:---:|:---:|:---:|:---:|
> |0|-942.91|0.0017|-0.1825|0.0076|3.3761|
> |1|-688.73|0.0026|-0.1905|0.0076|3.1313|
> |2|-602.39|0.0031|-0.2306|0.0074|2.9332|
> |3|-536.32|0.0032|-0.1853|0.0073|2.3901|
> |4|-489.57|0.0037|-0.2391|0.0072|2.8037|
> |5|-447.91|0.0041|-0.0671|0.0070|2.3695|
> |6|-410.26|0.0043|-0.0167|0.0070|2.5174|
> |7|-370.93|0.0044|0.0608|0.0069|2.1924|
> |8|-325.69|0.0047|0.1349|0.0067|1.5627|
> |9|-250.09|0.0049|0.0683|0.0063|1.1715|
>
> Results confirm our theory on what happens when fine-tuning a 7B model: when $y^-$ is highly unlikely (Buckets 0-6), DPO's effective gradient vanishes, pathologically decreasing the actual likelihood of the chosen response (Δchosen_sum < 0). Conversely, MIO maintains a robust gradient and consistently increases $y^+$ likelihood.
>
> ---
> **W4:Gains over DPO are small**
>
> **A4**: we conducted more evaluations of Mistral-7B-SFT trained with MIO, yields gains on Arena-Hard (+7.3 absolute, +44.7% relative over DPO).
>
> |Method|Arena-Hard|IFEval|BBH|GPQA|HumanEval|TheoremQA|LeetCode|
> |:--- |:---:|:---:|:---:|:---:|:---:|:---:|:---:|
> |DPO|16.3|43.29|43.27|28.35|31.7|9.8|2.2|
> |IPO|16.2|37.89|41.31|30.20|26.5|8.5|1.9|
> |MIO|**23.6**|**45.66**|**45.63**|**30.29**|**35.5**|**12.6**|**4.3**|
>
> In on-policy (TL;DR), Online MIO is also better：
>
> |Method|Win Rate|Quality|
> |:--- |:---:|:---:|
> |Online DPO|30.1|3.92|
> |Online IPO|28.9|3.86|
> |**Online MIO**| **35.5** | **4.01** |
>
> ---
> **Q1:Where is RLVR benchmarks?**
>
> **A5**: We agree that RLVR-style methods are relevant comparisons. However, our work focuses on RLHF, which relies on subjective preference signals rather than rule-based verification, so standard RLVR baselines are not directly applied in our setting. Still, we appreciate the broader question of how MIO compares with strong RL methods such as GRPO and GSPO.
>
> To address this, we adapted GRPO and GSPO to our setting by pairing them with an external reward model (Qwen2.5-1.5B-Instruct trained on ultrafeedback_binarized). Given rebuttal time constraints, we performed LoRA fine-tuning on Qwen2.5-7B-Instruct. As shown below, MIO outperforms these adapted baselines:
>
> |Method|HendrycksMath|MinervaMath|Multimedqa|MathQA|GSM8K|AquaRat|MathHard|MUSR|
> |:---|:---:|:---:|:---:|:---:|:---:|:---:|:---:|:---:|
> |GRPO|0.17|25.86|55.27|33.97|73.09|20.08|50.15|40.48|
> |GSPO|0.14|25.48|55.34|33.90|72.18|20.47|50.68|40.08|
> |**MIO**|**0.22**|**27.84**|**57.61**|**38.17**|**75.39**|**30.16**|**51.36**|**40.34**|

---

> > ### Author Rebuttal · Reviewer_RqEV · 2026-04-01
> >
> > I appreciate your response! I think the idea is good, however, the over claiming is a major issue as it guides the framing of the entire paper. Redoing the entire framing of the paper goes beyond an appropriate revision in my opinion.
> >
> > I recommend reframing fundamentally as a theoretical analysis that formalizes a connection between contrastive learning and DPO and develops an approach based on the observations of that connection. I would support this paper with that framing very strongly.

---

> > > ### Author Response · Authors · 2026-04-05
> > >
> > > We sincerely thank the reviewer for the continued discussion. We appreciate the opportunity to further clarify the intended scope of our contribution and theoretical framing.
> > >
> > > We agree that the original draft sometimes used overly strong language, and we fully accept the criticism that phrases such as **“strict equivalence,” “secret engine,” and “all along” were too broad**. In the final version, we will tone down these claims and revise the title and words accordingly, so that the paper is presented as **an approximate formal framework under explicit assumptions and bounded surrogate gaps**, rather than as an exact mathematical equivalence.
> > >
> > > At the same time, we would like to clarify that **our intended scope is broader than DPO alone**. Our goal is to analyze the connection between **RLHF-style preference optimization and contrastive learning**, including reward-model training (Eq. 8), the DPO loss (Eq. 10), and the standard KL-regularized RLHF objective (Eq. 11). Our claim is therefore not the broad statement that “DPO is contrastive learning.” Rather, our contribution is the **specific DV/MINE-based derivation that links KL-regularized RLHF—rather than DPO alone—to a contrastive-learning perspective**, and that in turn **motivates replacing the DV surrogate with a JSD surrogate**.
> > >
> > > Within this framework, our main point is that the **DV/MINE surrogate induces a shared optimization pathology across RLHF-style methods**. In on-policy settings, **rare but valuable responses receive overly weak learning signals even when sampled**. In off-policy settings, such as mid-to-late DPO training, **when the rejected-response probability becomes extremely small, the effective gradient for improving the chosen response decays rapidly**, leading to the pathological simultaneous degradation of both chosen and rejected likelihoods. Importantly, **this pathology is supported both theoretically and empirically**. Theoretically, *Theorem 2 (DV/MINE Starvation Theorem Pro)* and the gradient analysis in *Section 3.2* prove that DV-bound methods (like DPO) suffer from this gradient starvation. Empirically, as detailed in *our previous response to W3*, this phenomenon is not isolated to our toy example; we directly **observed it in real 7B-model training**. When the rejected-response probability becomes highly unlikely, the DV-style objective severely hinders further improvement of the chosen response, whereas JSD-style maintains a robust learning signal.
> > >
> > > We also wish to emphasize that the connection is **expressed through an inequality rather than an exact equality does not diminish the scope of the framework**. The connection is approximate, but not merely heuristic: **we provide theoretical analysis of the approximation errors and surrogate gaps**, and in the revision we will present these bounds more clearly in the main text. Thus, while we will narrow the wording to be academically precise, we believe the paper’s contribution remains the same: a formal but approximate RLHF/contrastive connection, an analysis of the DV-induced optimization pathology, and a JSD-based alternative motivated by that analysis.

---

### Official Review · Reviewer_Hsix · 2026-03-12

**Soundness:** 2
**Presentation:** 2
**Significance:** 2
**Originality:** 3
**Overall Recommendation:** 5
**Confidence:** 4

**Summary:**

This work deep dives into the theoretical connection of RLHF and DPO to contrastive learning, interpreting from the perspective of mutual information (MI) maximization.
Driven by this connection, this work reveals why RLHF cannot correctly update when the probability of sampling the correct answer from the base model is close to 0.
Accordingly, the authors propose to replace the Donsker-Varadhan lower bound on MI with Jensen-Shannon MI estimator.
Comprehensive theoretical analysis and extensive empirical evaluations demonstrate that the proposed technique mitigates the late-stage decline in chosen-likelihood observed in DPO, achieving competitive or superior performance across various challenging reasoning and mathematical benchmarks.

**Compliance With Llm Reviewing Policy:**

Affirmed.

**Final Justification:**

My concerns are addressed by the authors' responses.

**Key Questions For Authors:**

1. In `Answer to Question 2`, shall $\pi\_\theta(y^\star | x^\star)$ be $\pi\_\text{ref}(y^\star | x^\star)$? As most of the discussion is driven by $\pi\_\text{ref}(y^\star | x^\star) \to 0$.


2. Is MIO only applicable to DPO-like method with reference models? Can it improve PPO (no direct negative/positive pairs) or SimPO (no reference models)?

**Limitations:**

yes

**Strengths And Weaknesses:**

## Strengths

1. The paper is well motivated in its attempt to improve the understanding and effectiveness of RLHF-based alignment methods.
2. The theoretical connection between RLHF/DPO and contrastive learning is clearly stated and well developed.
3. The work provides comprehensive theoretical analysis along with extensive empirical evaluations to support the advantages of the proposed approach.


## Weaknesses

1. Even though I can get the high-level idea of the theoretical analysis (Sec. 2), it is a bit confusing when the scope is on general RLHF beyond DPO (including PPO and others).
    - 1). PPO is online setting with a single rollout each generation, while DPO is offline setting where trajectories are provided directly in the training data. There are some detailed differences between PPO and DPO, which make the theoretical analysis not hold for them simultaneously without careful discussion.
        - In a mini-batch, PPO might have all positive or all negative samples, which contrast the assumption of Eq (6).
        - In Sec 2.1, it highly relies on the specific case that $\pi_\text{ref}(y^\star | x^\star)$ approaches zero. However, this is highly unlikely for PPO case in the online setting.  (It is still valid for DPO).
    - 2). The analytic framework seems not applicable to RLHF techniques without reference model, such as SimPO (Meng et al., 2024). More in-depth discussion will strengthen this work.

> I believe the analysis is well suited for DPO-style RLHF techniques. However, when extending the discussion to PPO-style RLHF, additional clarification and consideration may be necessary.

2. This work has strong emphasis on the reasoning capacity enhancement, which is typically not the main focus of DPO.

3. In the math reasoning experiments, it may also be beneficial to include GRPO in the discussion and empirical comparison. A recent work (Wu et al., 2025) suggests that GRPO can be connected to DPO from a contrastive learning perspective in the context of math reasoning. Including a discussion and comparison with GRPO or Wu et al., (2025) could help better position the proposed method as well as potentially extend the scope of this work.



------
- Meng, Yu, et al. "Simpo: Simple preference optimization with a reference-free reward." Advances in Neural Information Processing Systems 37 (2024): 124198-124235.
- Wu, Yihong, et al. "It takes two: Your grpo is secretly dpo." arXiv preprint arXiv:2510.00977 (2025).

---

> ### Author Rebuttal · Authors · 2026-03-31
>
> Thank you for your insightful and valuable feedback.
>
> ---
> **W1: Distinguishing Online and Offline Settings**
>
> **A1:** We thank the reviewer for highlighting the need to distinguish between offline and online settings. We will explicitly clarify these boundaries in the revision.
>
> * **PPO-style RLHF & Eq. (6):** Our analysis is fundamentally on-policy (e.g., Eq. 11 samples $y \sim \pi_\theta(\cdot \mid x)$). We do not claim to model exact clipped PPO dynamics, nor do we assume every online mini-batch contains explicit positive/negative pairs (Eq. 6). Rather, Eq. 6 serves as a surrogate lens to explain how the underlying DV-bound induces gradient starvation.
> * **Online DPO:** Our core argument is that even if a rare, high-value response is successfully sampled, its extremely low policy probability still yields a weak learning signal, preventing the model from exploiting rare-but-correct responses.
> * **Different Manifestations:** We agree the failure mode manifests differently across settings. In **offline DPO**, DV starvation drives the late-stage synchronous collapse. In **PPO-style RLHF**, it explains the inherent difficulty in learning rare-but-correct responses (highlighting the practical necessity of SFT/cold-start initialization).
> ---
> **W2：More alignment benchmark**
> we conducted more evaluations of Mistral-7B-SFT trained with MIO, yields gains on Arena-Hard (+7.3 absolute, +44.7% relative over DPO).
>
> |Method|Arena-Hard|IFEval|BBH|GPQA|HumanEval|TheoremQA|LeetCode|
> |:--- |:---:|:---:|:---:|:---:|:---:|:---:|:---:|
> |DPO|16.3|43.29|43.27|28.35|31.7|9.8|2.2|
> |IPO|16.2|37.89|41.31|30.20|26.5|8.5|1.9|
> |MIO|**23.6**|**45.66**|**45.63**|**30.29**|**35.5**|**12.6**|**4.3**|
>
> In on-policy (TL;DR) on Qwen3.5-9B, Online MIO is also better：
>
> |Method|Win Rate|Quality|
> |:--- |:---:|:---:|
> |Online DPO|30.1|3.92|
> |Online IPO|28.9|3.86|
> |Online MIO| 35.5|4.01|
>
> ---
> **W3: Related Work**
>
> **A3**: We will gladly discuss *Wu et al. (2025)* in the revision. our framework can naturally interpret GRPO updates by treating positive/negative-advantage samples as chosen/reject pairs.
>
> To compare, we adapted GRPO and GSPO to our setting via an external reward model (Qwen2.5-1.5B-Instruct on UltraFeedback). Using LoRA on Qwen2.5-7B-Instruct (due to time constraints), we found **MIO still achieves the best performance** on almost all benchmarks:
>
> |Method|HendrycksMath|MinervaMath|Multimedqa|MathQA|GSM8K|AquaRat|MathHard|MUSR|
> |:---|:---:|:---:|:---:|:---:|:---:|:---:|:---:|:---:|
> |GRPO|0.17|25.86|55.27|33.97|73.09|20.08|50.15|**40.48**|
> |GSPO|0.14|25.48|55.34|33.90|72.18|20.47|50.68|40.08|
> |**MIO**|**0.22**|**27.84**|**57.61**|**38.17**|**75.39**|**30.16**|**51.36**|40.34|
>
> ---
> **Q1:shall be  $\pi_{\mathrm{ref}}(y^\*|x^\*)$,?**
>
> **A4**:The exact-zero boundary result is stated in terms of $\pi_{\mathrm{ref}}(y^\*|x^\*)$, since it relies on the support of the reference/base model under our reference-based setup. This explains why responses outside the effective support of the base model are hard to recover.
>
> By contrast, the practical near-zero starvation result is stated in terms of $\pi_\theta(y^\*|x^\*)$, because the effective update is governed by the **current policy**. This is especially important for on-policy PPO-style RLHF, where sampling and optimization are both driven by $\pi_\theta$, not by $\pi_{ref}$. Therefore, replacing $\pi_\theta$ with $\pi_{ref}$ would make the near-zero analysis less appropriate for the on-policy setting.
>
> We agree that the current draft may blur these two roles, and we will distinguish them more clearly.
>
> ---
> **Q2: Can MIO improve PPO or SimPO?**
> **A5:** For PPO batches lacking explicit negative samples, by applying intra-batch reward normalization (akin to GRPO's relative advantage), even an all-positive PPO batch naturally splits into relative positive and negative signals. Due to time constraints, we leave the full PPO validation to future work.
>
> For **SimPO (reference-free)**, our approach applies directly. The original loss relies on the pairwise difference $s^+ - s^-$ (where $s^\pm = \frac{\beta}{|y|} \log \pi_\theta(y^\pm|x)$). When the rejected response becomes extremely unlikely ($s^- \to -\infty$), the gradient on the chosen response still vanishes. To prevent this, we replace the single softplus with our **JSD-style three-term objective**:
>
> $$
> L_{\text{JSD-SimPO}}=\operatorname{sp}\left(-\left(s^+-\frac{\gamma}{2}\right)\right)+\frac12\operatorname{sp}\left(s^+-\frac{\gamma}{2}\right)+\frac12\operatorname{sp}\left(s^-+\frac{\gamma}{2}\right)
> $$
>
> Using LoRA fine-tuning on Qwen2.5-7B-Instruct, **JSD-SimPO consistently outperforms SimPO**, proving that our JSD formulation effectively generalizes beyond reference-based DPO:
>
> | Method | BBH | GPQA | IFEval | MATH | MMLU | MuSR |
> | :--- | :---: | :---: | :---: | :---: | :---: | :---: |
> | SimPO | 53.65 | 27.43 | 78.66 | 37.96 | 42.82 | 37.34 |
> | **JSD-SimPO** | **53.86** | **27.65** | **79.14** | **46.03** | **42.72** | **41.27** |

---

> > ### Author Rebuttal · Reviewer_Hsix · 2026-04-03
> >
> > Thanks for the responses.
> >
> > My concerns are addressed.
> >
> > I will update the score accordingly.

---

> > > ### Author Response · Authors · 2026-04-05
> > >
> > > We sincerely thank you for your thoughtful follow-up and for carefully reading our rebuttal.
> > >
> > > We truly appreciate your constructive feedback throughout the review process. Your comments were highly valuable in helping us clarify the scope of our analysis, especially regarding the distinction between offline DPO-style settings and on-policy PPO-style RLHF, as well as the applicability of our framework beyond reference-based methods. Your suggestions also motivated us to strengthen the discussion of related work and broaden the empirical positioning of our method.
> > >
> > > We are very grateful that our additional clarifications and new results were able to address your concerns. Thank you again for your careful evaluation, helpful feedback, and support.

---

### Official Review · Reviewer_7Fsp · 2026-03-12

**Soundness:** 3
**Presentation:** 2
**Significance:** 3
**Originality:** 3
**Overall Recommendation:** 5
**Confidence:** 4

**Summary:**

This paper proposes a unified mutual information (MI) perspective on RLHF and DPO, arguing that both can be interpreted as performing contrastive learning via the Donsker-Varadhan (DV) lower bound on MI. Under this perspective, reward learning corresponds to searching for a critic function that minimizes the discrepancy between estimated and actual mutual information, and policy optimization corresponds to maximizing MI between the policy and preferred responses while minimizing MI with rejected responses.

The paper identifies a key failure mode: the DV/MINE estimator yields vanishing gradients when the policy probability $\pi_\theta(y^{\star}|x^{\star})$ approaches zero (Theorems 1 and 2). This explains both why PPO/GRPO require cold-start initialization and why DPO exhibits ''synchronous collapse'' where both chosen and rejected response likelihoods decrease simultaneously. To address this, the paper replaces the DV estimator with the Jensen-Shannon (JS) MI estimator, yielding Mutual Information Optimization (MIO). MIO is shown theoretically and empirically to avoid the synchronous decline issue. Experiments on three base models (Mistral-7B-SFT, LLaMA3-8B-SFT, Qwen2.5-7B-Instruct) across eight reasoning and mathematical benchmarks demonstrate competitive or superior performance compared to DPO and its variants.

**Compliance With Llm Reviewing Policy:**

Affirmed.

**Final Justification:**

The authors have answered many of my concerns, and have decided to maintain my score.

**Key Questions For Authors:**

1. The MIO $\beta$ parameter (Table 4) appears to take very different values (5.0-10.0) compared to DPO's $\beta$ (0.01-0.1). Can you provide guidance on setting this parameter, and how sensitive is MIO to $\beta$ selection?
2. How does MIO perform when combined with techniques like rejection sampling or iterative DPO (online DPO)? The current experiments use offline preference data only.

**Limitations:**

The authors discuss the DV estimator limitations but do not explore alternative MI estimators beyond JS. The impact statement is thorough regarding potential misuse risks.

**Strengths And Weaknesses:**

## Soundness
* (**Strength**): The DV/MINE Starvation Theorems (Theorems 1 and 2) are rigorously stated and proven (Appendix H). The gradient analysis of MIO (Propositions 3.1 and 3.2) shows the self-regulating property: MIO's gradient on the chosen response flips sign based on $\sigma^+ > 2/3$ or $\sigma^+ < 2/3$.
* (**Weakness**): The derivation from RLHF to contrastive learning involves several approximations whose tightness might not have be adequately analyzed. For example, the authors introduce a mixture pool relaxation (Eqs. 3, 4, and 5) that introduces a known gap. While the authors address this in appendix C, further discussion on how much MI is lost due to the restriction could further strengthen the intuition that reward learning is "equivalent to" searching for a critic; currently, the expressiveness gap seems to limit this analogy. Additionally, the energy-based model assumption for $\pi_{\text{chosen}}$ (Appendix A, Eq. A.1) might be too strong of a structural assumption that may not hold in practice.

## Presentation
* (**Strength**): The toy model experiments (Figures 1 and 2) effectively illustrate the theoretical predictions about DPO collapse vs. MIO stability across different initialization regimes.
* (**Weakness**): Minor--The paper has a typo in the title (''Contarstive'' should be ''Contrastive'').

## Significance
* (**Strength**): The synchronous collapse of chosen and rejected likelihoods in DPO is a well-documented empirical phenomenon. Providing a principled theoretical explanation and a concrete algorithmic fix (MIO) is a valuable contribution to the field.
* (**Weakness**): The experimental improvements over DPO are modest on many benchmarks (Table 1). For example, on Hendrycks Math, the improvements are marginal (e.g., 0.21 vs. 0.04 for Mistral, but both are extremely low). The comparison to PPO (Table 3) shows MIO is slightly better than DPO but the setting (GPT-J on TL;DR) might be dated. Modern RLHF evaluations typically use larger models.

## Originality
* (**Strength**): The MI perspective on RLHF is insightful and provides a unifying view of reward learning, DPO, and policy optimization. The connection between the BT preference model and the DV/MINE estimator (Eqs. 7 and 8) is well derived. The identification of DV estimator starvation as the root cause of DPO's synchronous collapse is a novel theoretical contribution.
* (**Weakness**): The connection between preference learning and contrastive learning has been noted by prior work ([Xu et al., 2024] and [Chen et al., 2024]), and while this paper provides a more rigorous theoretical foundation, the conceptual insight is not entirely new. The replacement of DV with JS estimators follows directly from the mutual information estimation literature [Hjelm, R. D. et al., 2019].

---

> ### Author Rebuttal · Authors · 2026-03-31
>
> We deeply appreciate your insightful and valuable feedback.
>
> ---
> **W1:  Adequately analyzed of tightness**
>
> **A1**: We agree that the RLHF-to-contrastive-learning connection is exact at the unrestricted DV level, while the practical objective introduces a small and quantifiable surrogate gap. We will revise the wording accordingly and avoid overstating the claim as a strict equivalence.
>
> For the mixture-pool relaxation in Appendix C, let
>
> $$A_T = \mathbb E_{x, y^+ \sim \pi_{{chosen}}}[e^{T(x,y^+)}], \qquad B_T = \mathbb E_{x, y^- \sim \pi_{{rejection}}}[e^{T(x,y^-)}].$$
>
> For any fixed critic $T$, the gap is exactly
>
> $$\Delta_{{pool}}(T) = \log(A_T+B_T) - \log A_T = \log\left(1 + \frac{B_T}{A_T}\right) = \log(1 + e^{-m_T}),$$
>
> where $m_T = \log(A_T/B_T)$. Hence the error decays exponentially with the critic margin: if $m_T \ge 3$, then $\Delta_{\text{pool}} \le 4.9 \times 10^{-2}$; if $m_T \ge 4$, $\le 1.8 \times 10^{-2}$; and if $m_T \ge 5$, $\le 6.7 \times 10^{-3}$. So even a modest separation between chosen and rejected responses already makes this relaxation numerically very small.
>
> Equivalently, under the separated regime $P_Y^-(y) \le \rho P_Y^+(y)$, the marginal gap satisfies
>
> $$\Delta_{{mix}} \le \log(1+\rho),$$
>
> which gives $\Delta_{{mix}} \le 9.5 \times 10^{-2}$ for $\rho=0.1$, and $\le 4.9 \times 10^{-2}$ for $\rho=0.05$.
>
> The reviewer is also right that restricting the critic to the log-ratio family introduces an expressiveness gap, but this gap is second-order small. If $T^\star$ is the optimal unrestricted critic and there exists $T$ in the restricted family with $\|T-T^\star\|_\infty \le \varepsilon$, then
>
> $$\Delta_{{crit}} \le \frac{\varepsilon^2}{2}.$$
>
> Numerically, this gives $\Delta_{\text{crit}} \le 5 \times 10^{-3}$ for $\varepsilon=0.1$, and $\le 2 \times 10^{-2}$ for $\varepsilon=0.2$.
>
> Regarding Appendix A.1, we view it mainly as a density-ratio reparameterization rather than a strong additional assumption: if
>
> $$\text{supp}(\pi_{{chosen}}(\cdot|x)) \subseteq {supp}(\pi_{{ref}}(\cdot|x))$$
>
> then Eq. (A.1) is exact and introduces zero approximation error. If there is a small support mismatch of mass $\eta(x)$, its effect is only linear:
>
> $$|\mathbb E_{\pi_{\text{chosen}}}[f] - \mathbb E_{\tilde\pi_{\text{chosen}}}[f]| \le 2\|f\|_\infty \eta(x).$$
>
> Conceptually, $\eta(x)$ measures the probability weight of responses in $\pi_{\text{chosen}}$ that are deemed impossible by the reference model $\pi_{\text{ref}}$:
>
> $$\eta(x) := \pi_{{chosen}}(\{y : \pi_{{ref}}(y|x) = 0\} | x)$$
>
> So the real issue is support leakage, not the energy-based form itself.
>
> We will revise the paper to make this point clearer: the connection is exact at the unrestricted DV level, and the practical surrogate remains a close approximation in the regime relevant to preference optimization.
>
> ---
> **W2:Typo**
>
> **A2:** Thank you for pointing this out. We have corrected this typo as well as several other minor spelling and grammatical errors throughout the paper.
>
> ---
> **W3: Marginal improvements and dated comparison**
>
> **A3:We agree that the improvements in the original submission are modest on some benchmarks, and the GPT-J/TL;DR comparison is somewhat dated. To address this, we added a more modern online experiment on Qwen-3.5-9B, where **Online MIO** outperforms others:
> |Method|Win Rate|Quality|
> |:--- |:---:|:---:|
> |Online DPO|30.1|3.92|
> |Online IPO|28.9|3.86|
> |**Online MIO**|**35.5**|**4.01**|
>
> In addition, we conducted further evaluations on several stronger benchmarks for Mistral-7B-SFT, where MIO also shows consistent gains over DPO and IPO:
>
> |Method|Arena-Hard|IFEval|BBH|GPQA|HumanEval|TheoremQA|LeetCode|
> |:--- |:---:|:---:|:---:|:---:|:---:|:---:|:---:|
> |DPO|16.3|43.29|43.27|28.35|31.7|9.8|2.2|
> |IPO|16.2|37.89|41.31|30.20|26.5|8.5|1.9|
> |MIO|**23.6**|**45.66**|**45.63**|**30.29**|**35.5**|**12.6**|**4.3**|
>
> ---
> **Q1: Parameter setting guidance and sensitivity analysis**
>
> **A4**: The $\beta$ values are not directly comparable across different loss functions. Our larger $\beta$ naturally results from using average token log-probability normalization (in Appendix G, following TRL's IPO/SimPO/DIL, where SimPO also uses a larger default $\beta=2.5$).
>
> Empirically, MIO is reasonably stable over a broad range of $β$ values. In our added sensitivity experiment, performance remains competitive throughout the 2.0–12.5 range: smaller $β$ values slightly favor instruction-following, while moderately larger $β$ values slightly favor reasoning.
>
> |Metric/beta|2.0|5.0|7.5|10.0|12.5|15.0|20.0|
> |:---|:---:|:---:|:---:|:---:|:---:|:---:|:---:|
> |IFEval|46.37|45.96|45.88|45.66|43.38|39.19|36.59|
> |GSM8K|39.91|39.35|39.33|40.16|39.77|39.57|38.74|
>
> ---
> **Q2: Combined with on policy methods**
>
> **A5**: MIO is compatible with online preference-optimization methods (detailed in A3). These results suggest that the benefit of MIO is not limited to offline preference data, and that it remains effective in an online setting as well.

---

> > ### Author Rebuttal · Reviewer_7Fsp · 2026-04-04
> >
> > The authors have fully answered my questions and provided helpful further evidence.

---

> > > ### Author Response · Authors · 2026-04-05
> > >
> > > We sincerely thank you for your thoughtful follow-up and for carefully reading our rebuttal.
> > >
> > > We greatly appreciate your recognition that our additional clarifications on the approximation gap and modeling assumptions, the corrected presentation issues, and the newly added empirical evidence have adequately addressed your concerns. Your feedback was highly valuable in helping us sharpen the paper, especially in making the theoretical claims more precise, improving the presentation, and strengthening the experimental section with more modern online results and parameter sensitivity analysis.
> > >
> > > We are especially grateful for your constructive and insightful comments throughout the review process. Your suggestions directly contributed to improving both the clarity and the completeness of the manuscript, and we are very pleased that our rebuttal was able to fully answer your questions.
> > >
> > > Thank you again for your careful evaluation, helpful feedback, and support.

---

### Official Review · Reviewer_UH3r · 2026-03-13

**Soundness:** 3
**Presentation:** 3
**Significance:** 3
**Originality:** 3
**Overall Recommendation:** 5
**Confidence:** 3

**Summary:**

The paper presents a unified take on the contrastive loss inherent in both RLHF and DPO, in terms of a lower bound on mutual information between the policy & preferred response distribution. It presents  explanations of why this leads to vanishing gradients when reference policy probs are close to 0, and why chosen and rejected probs reduce in DPO. The authors propose an alternative algorithm, MIO, that replaces the lower bound with a Jensen-Shannon MI estimator & show improved empirical results on benchmarks compared to DPO and variants.

**Compliance With Llm Reviewing Policy:**

Affirmed.

**Final Justification:**

The paper offers a useful and interesting alternative to preference optimisation algorithms like DPO and its variants called "MIO" based on mutual information. The authors' technique is theoretically well motivated based on the relationship between KL-regularised RLHF and contrastive learning, and offers a principled alternative in situations where DPO suffers gradient pathologies. Indeed, their technique is borne out in both offline & online settings (new experiments in rebuttal). Overall this is a valuable piece of work that will be valuable to the field of preference tuning & RLHF.

**Key Questions For Authors:**

- Why is it desirable to overcome the close-to-zero policy prob behaviour of previous techniques? may actually prevent catastrophic forgetting, are there any tradeoffs of doing so with MIO?
- Why does MIO use betas that are so massive compared to DPO? maybe i am misunderstanding the parameter
- Do chosen rewards actually go up, esp. compared to techniques like iterative RPO that address this explicitly?

**Limitations:**

yes

**Strengths And Weaknesses:**

The paper offers MIO, which is an interesting alternative to preference optimisation algorithms like DPO and its variants - the mutual information based loss is definitely an interesting comparison to KL based losses. The benchmark numbers seem to check out with small wins for MIO, promising on math/reasoning benchmarks (although here on-policy RL is usually stronger & preferred compared to off-policy), whereas wins are small on alignment benchmarks that are quite relevant to the use of DPO.

However many claims about theoretical novelty, unification & some results are inflated, unsupported or poorly motivated - for instance chosen rewards seem to barely increase with MIO (and this is better addressed by hybrid losses e.g. the iterative RPO paper), and the unification framing is misleading - DPO is known to be a contrastive technique, off-policy-ness is already quite well understood (and penalising it is generally thought of as desirable), and traditional RLHF/PPO is importantly *on-policy* (which makes a big difference).

---

> ### Author Rebuttal · Authors · 2026-03-31
>
> We sincerely thank the reviewer for the constructive feedback and the detailed examination of our work.
>
> ---
> **W1:  The benchmark numbers seem to small wins for MIO**
>
> **A1**: To address your concern regarding the magnitude of the improvements, we conducted comprehensive evaluations of Mistral-7B-SFT trained with MIO, yields gains on Arena-Hard (+7.3 absolute, +44.7% relative over DPO).
>
> |Method|Arena-Hard|IFEval|BBH|GPQA|HumanEval|TheoremQA|LeetCode|
> |:--- |:---:|:---:|:---:|:---:|:---:|:---:|:---:|
> |DPO|16.3|43.29|43.27|28.35|31.7|9.8|2.2|
> |IPO|16.2|37.89|41.31|30.20|26.5|8.5|1.9|
> |MIO|**23.6**|**45.66**|**45.63**|**30.29**|**35.5**|**12.6**|**4.3**|
>
> In online RLHF (TL;DR), Online MIO outperforms others (detailed in A4). We will add these to the revision.
>
> ---
> **W2:  the unification framing is misleading - DPO is known to be a contrastive technique, off-policy-ness is already quite well understood**
>
> **A2**: We sincerely apologize if the wording in the original draft came across as overly broad or misleading. We have taken your feedback to heart and changed the title of our paper to: **"The Hidden Link between RLHF and Contrastive Learning"**. We also commit to revising the framing in the main text to be more precise.
>
> To clarify, our novelty is not the broad statement that "DPO is contrastive" or that off-policy issues exist. Rather, **it is the specific DV/MINE-based derivation** that links KL-regularized RLHF and rigorously motivates the JSD replacement. Our core claim is that MIO offers a consistent, theoretically motivated alternative that is uniquely beneficial in settings where DPO’s gradient pathology (vanishing gradients for chosen responses) becomes problematic.
>
> ---
> **W3&Q3: Do chosen rewards actually go up compare to IRPO?**
>
> Yes, MIO increases chosen likelihood without simultaneously raising rejected likelihood. Unlike IRPO—which forces chosen probability up but causes empirical performance degradation—MIO’s naturally bounded JSD surrogate avoids these negative side effects. See dynamics: [Link](https://i.postimg.cc/1RvtqCVy/MIO-IRPO.png)
>
> ---
> **W4: DPO is known to be a off-policy contrastive technique and traditional RLHF/PPO is importantly on-policy**
>
> **A4**: While our DPO discussion focuses on off-policy scenarios, our PPO analysis is fundamentally on-policy. Specifically, Eq. (11) is explicitly on-policy as it samples $y \sim \pi_\theta(\cdot|x)$. Our starvation theorem applies to the RLHF sampling scheme (covering both reward-model and log-ratio critics) and explains the failure mode via the DV/MINE surrogate underlying PPO-style RLHF, without claiming to model exact clipped PPO dynamics.Importantly, MIO seamlessly supports on-policy training. In our newly conducted online RLHF experiment (training Qwen-3.5-9B on TL;DR with DeepSeek-V3.2 as the judge), Online MIO outperforms both Online DPO and IPO:
>
> |Method|Win Rate|Quality|
> |:--- |:---:|:---:|
> |Online DPO|30.1|3.92|
> |Online IPO|28.9|3.86|
> |**Online MIO**| **35.5** | **4.01** |
>
> ---
> **Q1.1: Why to overcome the "near-zero policy probability behavior"?**
>
> **A5**: While some conservative effects of low probability are acceptable, DV/MINE-based starvation suppresses learning from highly informative comparisons, acting as a hindrance rather than desirable regularization.
>
> On-policy scenarios: Low-probability, high-value responses yield weak learning signals even when sampled. This prevents the model from effectively utilizing them, which necessitates "cold-start" phases in standard RLHF.
>
> Off-policy scenarios (especially mid-to-late DPO): As the rejected response probability approaches zero, DPO’s effective gradient for the chosen response decays exponentially. This causes a pathological simultaneous decrease in both chosen and rejected likelihoods.
>
> ---
> **Q1.2: Are there any tradeoffs of doing so with MIO?**
>
> **A6**:Yes. MIO's stronger learning signal makes it slightly more sensitive to label noise. To quantify this, we trained Qwen-2.5-7B-Instruct on the UltraFeedback dataset with 10% randomly flipped preference labels.
>
> |Method|MUSR|
> |:---|:---:|
> |Qwen|38.5|
> |DPO|36.7|
> |MIO|36.0|
>
> Because MIO extracts more signal from the given labels, it experiences a slightly larger performance drop on MUSR when exposed to corrupted data compared to DPO. **This confirms that MIO's efficacy relies on reasonably high-quality preference data**.
>
> ---
> **Q2：MIO use beta so massive compared to DPO?**
>
> **A7**:$\beta$ values aren't directly comparable between objectives due to fundamentally different loss functions. Empirically, MIO remains highly stable across a broad $\beta$ range:
>
> |Metric/beta|2.0|5.0|7.5|10.0|12.5|15.0|20.0|
> |:---|:---:|:---:|:---:|:---:|:---:|:---:|:---:|
> |IFEval|46.37|45.96|45.88|45.66|43.38|39.19|36.59|
> |GSM8K|39.91|39.35|39.33|40.16|39.77|39.57|38.74|
>
> MIO performs consistently well within the 2.0–12.5 range. Smaller $\beta$ favors instruction-following (IFEval), while slightly larger $\beta$ modestly improves reasoning (GSM8K).

---

> > ### Author Rebuttal · Reviewer_UH3r · 2026-04-01
> >
> > The authors have thoughtfully and comprehensively acknowledged all my concerns - the more specific framing, the new online RLHF experiments and the discussion of tradeoffs (esp. the sensitivity to pref. quality) make for a substantially better manuscript and a worthy contribution to the field. Kudos to the authors for taking the time and effort to respond in such detail!

---

> > > ### Author Response · Authors · 2026-04-05
> > >
> > > We sincerely thank you for your thoughtful follow-up and for taking the time to carefully read our rebuttal.
> > >
> > > We truly appreciate your recognition that our additional experiments on increasing the chosen likelihood, the new online RLHF results, and the discussion of tradeoffs have substantially strengthened the paper. Your earlier comments were highly valuable in helping us clarify the scope of our claims, improve the presentation of our contributions, and better articulate both the strengths and limitations of MIO.
> > >
> > > We are especially grateful for your constructive and detailed feedback throughout the review process. Your suggestions directly helped us improve the manuscript, and we are pleased that our rebuttal was able to address your concerns.
> > >
> > > Thank you again for your time, careful evaluation, and encouragement.

---

### Decision · Program_Chairs · 2026-04-30

**Decision:**

Accept (regular)

**Comment:**

This paper makes a strong contribution on preference optimization and RLHF. The submission offers a useful mutual-information-based perspective connecting RLHF/DPO and contrastive learning, identifies a concrete gradient-starvation failure mode for DV-style objectives, and proposes MIO as a targeted fix based on a JSD estimator. Three reviewers were strongly positive, and the rebuttal materially improved the paper by narrowing overclaims, adding stronger modern benchmarks, providing online RLHF evidence, and discussing tradeoffs such as sensitivity to label noise.

I agree with the main positive assessment. The paper is not claiming that the general connection between preference optimization and contrastive learning is entirely new; rather, its value is in making that connection precise enough to motivate a practical fix. I also agree with the remaining critical reviewer that the final version should stay disciplined about framing approximate connections as approximate, and should present MIO as a focused estimator substitution rather than a wholesale new paradigm. But I do not think those framing issues outweigh the strengths of the submission.

Overall, this is a solid ICML paper with a useful combination of theory, mechanism analysis, and empirical evidence. I recommend acceptance.